# A Higher Lignin Content in *ugt72b37* Poplar Mutants Indicates a Role of Monolignol Glycosylation in Xylem Lignification

Hadjara Amadou Hassane [1,2], Marc Behr [1], Claire Guérin [1], Richard Sibout [3], Adeline Mol [1], Moussa Baragé [2], Mondher El Jaziri [1] and Marie Baucher [1,*]

1   Laboratoire de Biotechnologie Végétale, Université libre de Bruxelles, Rue des Professeurs Jeener et Brachet 12, 6041 Gosselies, Belgium
2   Laboratoire de Biotechnologie et Amélioration des Plantes, Université Abdou-Moumouni, BP 10 727, Niamey 8004, Niger
3   UR1268 Biopolymères Interactions Assemblages, INRAE, 44300 Nantes, France
*   Correspondence: marie.baucher@ulb.be

**Abstract:** Plant UDP-glycosyltransferases (UGT) transfer sugars to small acceptor molecules and thereby play key roles in the biosynthesis of secondary metabolites, including phenylpropanoids. Some of those metabolites are involved in the xylem lignification of a broad range of terrestrial plants, particularly trees. Here, we focused on poplar *UGT72B37*, coding for an enzyme glycosylating monolignols by investigating CRISPR/Cas9 mutant lines. The cell wall characterization revealed a 10% lignin content increase in the xylem of three-month-old mutant lines compared to the wild type. No ectopic lignification was evidenced in the pith of the stems of the mutants, suggesting that the increased lignin deposition is restricted to lignified cell walls. The analysis of the expression level of lignin biosynthesis and polymerization genes did not show significant changes between the WT and the *ugt72b37* mutants, except for *CINNAMOYL-COA REDUCTASE 2* which was significantly upregulated by 1.2–1.5-fold. Noticeably, *UGT72B38*, the closest related gene to *UGT72B37*, is upregulated in mutant lines, suggesting a functional compensation between UGT72B37 and UGT72B38 possibly linked with lignin biosynthesis and accumulation in poplar. Overall, these results reinforce a plausible role of monolignol glycosylation in the cell wall lignification process.

**Keywords:** UGT72; poplar; lignin content; monolignol; xylem

## 1. Introduction

Lignin is a complex heteropolymer that is deposited in the cell walls of specialized cell types of vascular plants such as tracheids, vessels, fibers, sclerenchyma, and endodermis, thereby conferring mechanical support to cell walls and hydrophobicity to conducting cells [1]. Lignin derives mainly from three monolignols, the *p*-coumaryl, the coniferyl, and the sinapyl alcohols. These precursors, differing in the degree of methoxylation of their aromatic cycle, lead to the production of *p*-hydroxyphenyl (H), guaiacyl (G), and syringyl (S) units, respectively, when incorporated into the lignin polymer. Monolignol biosynthesis has been largely reviewed [2–6]. Following their biosynthesis in the cytoplasm, monolignols are transported to the cell wall [7–10], where they are oxidized by laccases (LACs) using $O_2$ and peroxidases (PRXs) using $H_2O_2$, and polymerized into lignin via a radical coupling process [11,12].

Monolignol 4-*O* position glycosides, i.e., *p*-coumaryl alcohol glucoside, coniferin, and syringin, are considered either as a storage or a transport form of *p*-coumaryl alcohol, coniferyl alcohol, and sinapyl alcohol, respectively [13]. Glycosylation of the monolignol aromatic ring increases their solubility and decreases their toxicity and reactivity [14]. An ATP-dependent transport activity for glycosylated monolignols through the vacuolar membrane, where they are stored [15], has been demonstrated for several plant species [16–20]. Monolignol glycosides have been found to accumulate in cambial tissue and differentiating

xylem in conifers and angiosperms [21–27]. The release of monolignols from their glucosidic form is catalyzed by cell wall coniferin-β-glucosidases [28,29]. The role of monolignol glycosides as lignin precursors is not clearly defined and is still a matter of debate. However, injection of radio-labeled glycosylated monolignols in stems of gymnosperms and angiosperms [30–35], as well as in vitro PRX-catalyzed dehydrogenative polymerization of coniferyl alcohol in the presence of monolignol glucosides [36], revealed that they are incorporated into the lignin polymer.

Several reports have shown that various UDP-glycosyltransferases (UGT) from the UGT72 family glycosylate hydroxycinnamic acids, cinnamaldehydes, cinnamyl alcohols as well as lignans with different affinities, such as the *Arabidopsis* UGT72E1-E3, UGT72B1 and UGT72B3 [37–42], the *Camellia sinensis* UGT72AM1 [43], the *Solanum lycopersicum* SlUGT5 [44], the *Vanilla planifolia* UGT72U1 [45], the *Isatis indigotica* IiUGT1 [46], the *Vitis vinifera* UGT7227 [47], or the *Pyrus bretschneideri* UGT72AJ2 [48]. In poplar (*Populus tremula × P. alba*), UGT72AZ2 glycosylates both ferulic and sinapic acids, UGT72B39 coniferyl alcohol, and UGT72B37, both coniferyl and sinapyl alcohols as well as *p*-coumaraldehyde, coniferaldehyde and sinapaldehyde [49]. Investigations of *Arabidopsis* and poplar transgenic lines with altered expression of members of the *UGT72* family have evidenced their role in the maintenance of the monolignols/monolignol glucosides homeostasis [42,50].

Particularly, the *Arabidopsis ugt72b1* mutant was found to have an ectopic lignification in interfascicular fibers and pith as evidenced in cross-sections treated for Wiesner, Mäule, and toluidine blue-O stainings [41]. Recently, the mutation of *LuUGT715*, the closest flax (*Linum usitatissimum)* homolog to *UGT72B1*, resulted also in reduced growth and an ectopic lignification within the stem [51]. The present study investigated the function of the poplar *UGT72B37*, which is homologous to the *Arabidopsis UGT72B1-B3* [49]. Poplar knockout mutants for *UGT72B37* were produced by Clustered Regularly Interspaced Short Palindromic Repeat (CRISPR)/CRISPR-associated protein 9 (Cas9) technology and the impact of this mutation on the lignification parameters of three-month-old plants was considered. In parallel, transgenic poplars overexpressing *UGT72B37* that were formerly produced [49], were also analyzed.

## 2. Materials and Methods

### 2.1. Protein Characterization

Protein modeling was realized using the AlphaFold tool [52] and visualized with PyMOL [53]. The UGT72B37 protein weighs 52.47 kDa and its isoelectric point is at pH 5.79, as calculated using the SMS2 tool [54]. Subcellular localization was investigated using Cell-PLoc 2.0 [55] and the Phyre2 tools [56]. According to the Phyre2 tool, UGT72B37 is constituted of α-helixes (43% of the protein sequence), β-sheet (13%), transmembrane helixes (10%), and the rest (16%) is labeled as disordered. The SMART database [57] was used to predict protein domain and structure. The NLS prediction was realized using the NLS mapper tool [58].

### 2.2. CRISPR/Cas9 sgRNA Design and Vector Construction

The *UGT72B37* (Potri.014G096100) gene has no intron, and the single guide RNA (sgRNA) GGCCACAACTCTAGTACCAC was designed using the genome of *Populus trichocarpa* and the e-CRISP tool [59] to target a site between c.323 and c.342. The sgRNA specificity was verified against the genome of *P. tremula × P. alba* on the Aspen DataBase portal (www.aspendb.org (accessed on 13 January 2020)). The targeted region was conserved in both alleles and specific to *UGT72B37*. The sgRNA was located in the corresponding protein sequence between the first NLS domain and the first transmembrane domain to maximize the probability of obtaining a loss-of-function protein.

The CRISPR/Cas9 construct was made using the p201N-Cas9 vector [60]. Multiple fragments and double-digested (SpeI and SwaI) p201N-Cas9 vector were Gibson-assembled using NEBuilder HiFi DNA assembly master mix (New England Biolabs, Ipswich, MA, USA). The assembled construct was verified by Sanger sequencing, then integrated into

*Agrobacterium tumefaciens* strain C58C1$^{rif}$ PMP90 by electroporation to perform a plant stable transformation.

### 2.3. Plant Material, Transformation, Mutation Screening, and Growth Conditions

*P. tremula* × *P. alba* stable transformation and selection (kanamycin) were performed as described previously [61]. Following the regeneration step, genomic DNA was extracted from the aerial part of vitroplants of 18 independent lines on which was performed high-fidelity PCR amplification of the genomic region targeted by the sgRNA (primers given in Table S1 in Supplementary Materials). Amplified fragments from six selected lines were cloned with the Zero Blunt TOPO PCR cloning kit (Invitrogen) and for each line, 10 independent colonies were Sanger-sequenced to identify the editing pattern as compared to the WT. Three independent lines carrying a knockout mutation in *UGT72B37* were further in vitro propagated and transferred to soil in a phytotron (photoperiod, 16 h/8 h; 22–24 °C; humidity, 30%–60%; light intensity, ca. 70 μmol m$^{-2}$ s$^{-1}$) for 3 months before sampling. The number of biological replicates, given in each analysis, corresponds to the number of individual trees per independent line.

Transgenic poplar lines overexpressing *UGT72B37* have been previously described [49].

### 2.4. Microscopy

A stem portion was sampled at 1, 10, 20, and 30 cm down to the apex, fixed in 70% ethanol, cut with a vibratome, and stained for 5 min with phloroglucinol (3% *w/v*) before HCl (12% *v/v*) incubation and observation with a light microscope.

### 2.5. Cell Wall Analysis, Lignin Composition, and Saccharification

Lignin content was determined using the cysteine-assisted sulfuric acid (CASA) method [62] on the 30 cm basal portion of the stem wood (approximately a piece at 70 cm below the apex). Dry wood was first ground to a fine powder using a Retsch MM 400 miller (5 min, 30 Hz). The wood powder (500 mg) was extracted thrice with 5 mL of ethanol 80% and once with 5 mL of pure acetone, yielding a cell wall residue (CWR) free of extractives, which was dried. Then, 10 mg of CWR was fully dissolved with 1 mL of 10% L-cysteine (*w/v*) in 72% sulfuric acid (1 h at room temperature under agitation). The solution was diluted to 100 mL with pure water and CASA lignin content was evaluated based on the absorbance at 283 nm and an absorption coefficient of 11.23 g$^{-1}$·L·cm$^{-1}$ [62]. The quantification of β-O-4 linked lignin units was assayed by thioacidolysis, as described in [63] and analyzed with an Agilent 5973 Gas Chromatography-Mass Spectrometry system (Santa Clara, USA).

Saccharification was performed on 10 mg wood powder pre-treated with mild acid condition (2 h at 70 °C in 1 mL 1M HCl). Pre-treated samples were washed thrice with 1 mL pure water and once with 1 mL of ethanol 70% (20 h incubation at 55 °C). The pellet was rinsed three additional times with 1 mL of ethanol 70%, once with 1 mL of pure acetone, and dried overnight at 50 °C. After weighing the tubes, the residue was dissolved in 1 mL acetate buffer (0.1 M, pH 5.1) and equilibrated at 50 °C for 5 min before adding 100 μL cellulase (Cellic CTec2, Novozymes, Bagsværd, Danemark). Digestion was performed at 50 °C and 5 μL aliquots were sampled at different time points. Aliquots were diluted 100 times in acetate buffer and the reaction was stopped by heating at 100 °C for 5 min. Released glucose was determined enzymatically by using the D-glucose GOPOD kit (Megazyme, Wicklow, Ireland) with 50 μL of sample solution and 1.5 mL of GOPOD reagent (incubation at 45 °C for 20 min), following the manufacturer's instructions.

### 2.6. Gene Expression Analysis

Debarked wood pieces (between 20 and 25 cm below the apex) were collected from the WT and two mutant lines (L6 and L16), then conserved at −80 °C and ground under liquid nitrogen. Total RNA was extracted using the Plant RNA Isolation Mini Kit (Agilent), according to the manufacturer's instructions. Quality-checked RNAs were retrotranscribed

using the ProtoScript II First Strand cDNA Synthesis Kit with the random primer mix (New England BioLabs, Ipswich, UK). Quantitative PCR was performed with 5 µL of diluted cDNA (1 ng·µL$^{-1}$), 0.5 µL of both primers (10 µM) (Table S1), 4 µL of water and 10 µL of Luna Universal qPCR Master Mix (New England BioLabs). The reactions were set up in a 96-well microplate, with technical triplicate, in a LightCycler 480 system (Roche, Bâle, Switzerland). Water control was performed for each primer pair. The qPCR program was composed of four steps: pre-incubation (95 °C, 1 min), amplification (95 °C, 15 s; 60 °C, 30 s, repeated 40 times), melting curve (95 °C, 5 s; 65 °C, 1 min; +0.1 °C/s until 97 °C) and cooling (40 °C, 10 min). The specificity of the amplification was assessed for each reaction with the melting curve. Relative quantities were computed using each primer pair's specific amplification efficiency (as previously determined with a standard curve) and normalized with two reference genes (*CDC2* and *CYC063*; [64]) by using the formula provided in [65]. WT as well as L6 and L16 mutant lines were analyzed in biological quadruplicates.

The expression of the following target genes was assessed: *PAL2* (Potri.008G038200), *C4H1* (Potri.013G157900), *C4H2* (Potri.019G130700), *4CL3* (Potri.001G036900), *CCR2* (Potri.003G181400), *HCT1* (Potri.003G183900), *CCoAOMT1/2* (Potri.009G099800/Potri.001G 304800), *F5H1* (Potri.005G117500), *F5H2* (Potri.007G016400), *COMT2* (Potri.012G006400), *CAD* (Potri.009G095800), *UGT72B36* (Potri.014G096000), *UGT72B38* (Potri.003G138200), *UGT72B39* (Potri.002G168600), *LAC4* (Potri.016G112100), *LAC17* (Potri.006G087100), *PRX72* (Potri.005G118700). The expression level of *UGT72AZ1* (Potri.007G030300) and *UGT72AZ2* (Potri.007G030400) was below the limit of quantification in all the samples.

## 3. Results

### 3.1. UGT72B37 Protein Characterization

UGT72B37-GFP was previously localized both in the nucleus and associated with the endoplasmic reticulum [49]. The modeled UGT72B37 protein is shown in Figure 1. Two transmembrane domains were identified, at aa 120–150 and aa 371–386, respectively, suggesting that UGT72B37 is a membrane-anchored protein. Besides, two bipartite nuclear localization signals (NLS) (at aa 26–54 and aa 202–229) were predicted with scores of 5.6 and 5.8, respectively.

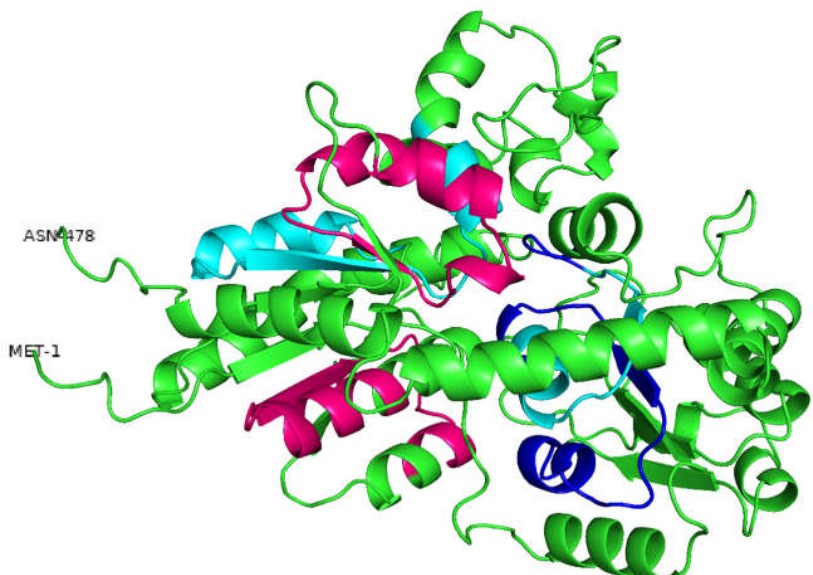

**Figure 1.** Three-dimensional structure of UGT72B37, modeled with AlphaFold tool and visualized with PyMOL. Light blue: transmembrane domain; dark blue: PSPG motif; red: bipartite NLS.

The UGT domain was identified between aa 254–444 (Pfam code PF00201, e-value $1.2 \times 10^{-18}$). This domain includes the Plant Secondary Product Glycosyltransferase (PSPG), the 44 amino acid C-terminal signature of the UGT family motif [66], at aa 346–390.

### 3.2. Generation of ugt72b37 Mutants via CRISPR/Cas9

In order to evaluate the role of UGT72B37 in lignification, a functional genomic approach was investigated by generating stable transgenic poplar mutants via CRISPR/Cas9. A single guide RNA (sgRNA) was designed to target a site located in the 5′-end of the coding region (Figure 2A) and mutated biallelic sequences were identified for three independent lines. Among these, L11 and L16 have biallelic frameshift mutations at c.325 position whereas L6 is heterozygous for the mutation showing a frameshift mutation for the *P. alba* allele and a codon deletion for the *P. tremula* allele at the target site (Figure 2B). The L6 mutant has a truncated protein of 200 aa for the *P. alba* allele, and a protein lacking one amino acid (a glycine) for the *P. tremula* allele. The length of the deduced polypeptides from the WT and the *ugt72b37* mutants is given in Figure 2C.

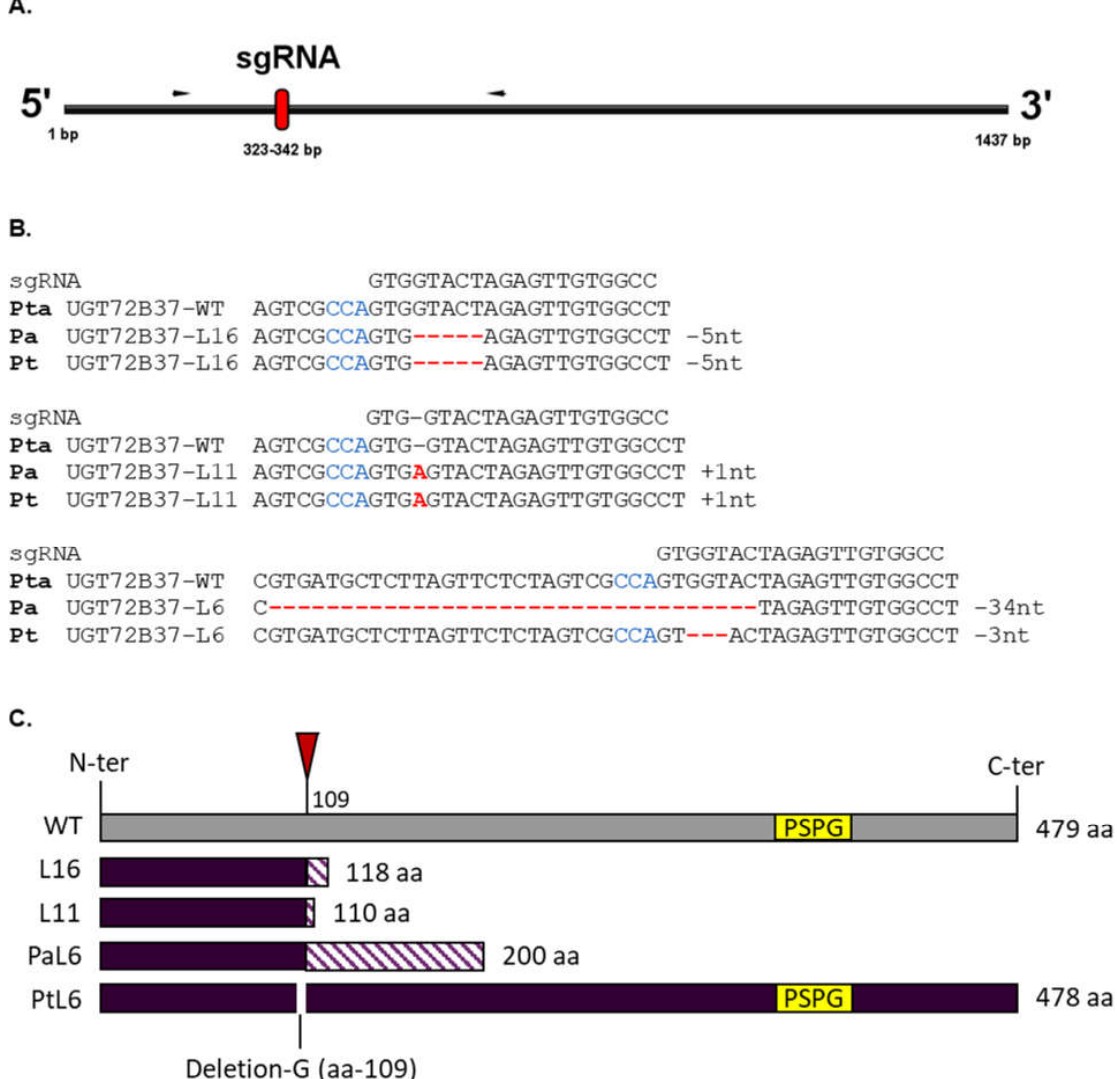

**Figure 2.** Production of transgenic poplars with CRISPR/Cas9-knockout of *UGT72B37*: (**A**) location of the sgRNA target site and the amplified region for sequencing (black arrows, 507 bp in the WT); (**B**) genotypes of the *ugt72b37* mutants used in this study. For each independent mutant line (L16, L11, and L6), the genomic modifications of the *P. alba* (Pa) and the *P. tremula* (Pt) alleles are indicated in bold red. The protospacer adjacent motif sequence is indicated in blue. The indel numbers are indicated on the right; and (**C**) length of deduced polypeptides from the WT and *ugt72b37* mutant lines. The PSPG motif is highlighted in yellow. The red motives in A and C indicate the sgRNA target site. The purple stripes represent a polypeptide sequence with a frameshift.

### 3.3. The ugt72b37 Mutation Triggers an Increased Lignin Content in Stem Xylem

*ugt72b37* mutants and WT were grown in a greenhouse for three months until they reached ~1 m high. No significant differences were observed in the overall phenotype of the mutants compared to the WT (Table S2). The impact of the *UGT72B37* mutation on lignin content was evaluated in stem tissues, where the gene was found mostly expressed, as analyzed by RT-qPCR and GUS staining in different plant organs and tissues [49]. The lignin content of dried debarked wood was measured, using the CASA method, on the 30 cm basal portion of the stem wood (~70 cm below the apex), which was made mostly of lignified secondary xylem. As shown in Figure 3, a significant increase of ~10% in lignin content was observed in the three mutant lines comparatively with the WT. The obtained values were 23.4%, 24.1%, and 23.7% for L16, L11, and L6 mutant lines, respectively, and 21.3% for WT. In order to evaluate whether the increased lignin content was caused by ectopic lignification, as previously reported for the *ugt72b1 Arabidopsis* mutant [41] and the *luugt715* flax mutant [51], transversal cross-sections were made along the stem of WT and the L16 *ugt72b37* mutant at 1, 10, 20, and 30 cm below the apex, to explore both primary and secondary structures. As shown in Figure 4, although not quantitative, the Wiesner staining seems stronger in the xylem of the stems of the mutant reinforcing the hypothesis that the increased lignin level in the *ugt72b37* mutants is associated with xylem cell walls. However, as poplars were in the growing stage and the annual ring was not fulfilled, these observations may indicate an early lignification in the *ugt72b37* mutant as it was shown in young stems of the *ugt72e3 Arabidopsis* mutant [67].

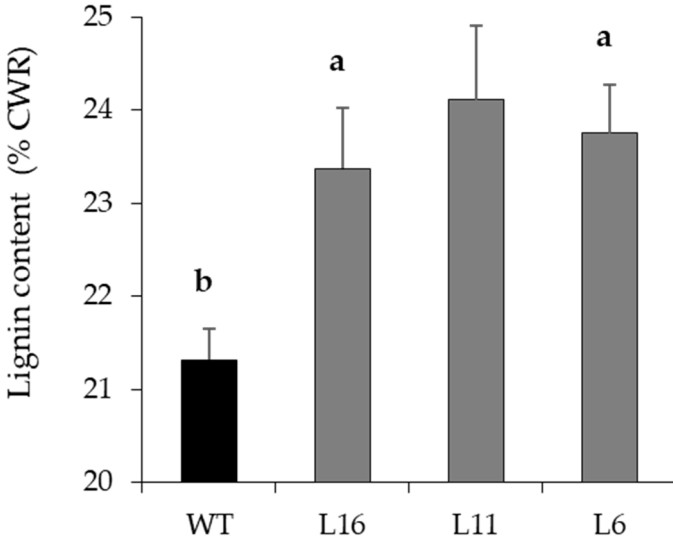

**Figure 3.** Lignin content of CWR in stems of three-month-old *ugt72b37* mutants and WT, determined by CASA method. Values are means of 5, 5, 4, and 2 biological replicates for WT, L16, L6, and L11, respectively, with two technical replicates. Error bars indicate SE. An ANOVA was performed (except for L11 because of a low number of biological replicates), and each letter specifies a statistically different group (*p*-value = 0.017).

Even though the mutation increased the total amount of lignin in the stem, it did not affect the percentage of H, G, and S units determined by thioacidolysis, nor the cellulose content and the saccharification efficiency of the CWR of the *ugt72b37* mutants as compared to the WT (Figure S1). However, when expressed with respect to lignin quantity, G and S unit contents were significantly lower in the *ugt72b37* mutant lines when compared to the WT (Figure S1), indicating a more condensed lignin, as the thioacidolysis procedure provides H, G and S thioethylated monomers from H, G and S units that were exclusively involved in arylglycerol-β-ether structures. In parallel, four-month-old transgenic poplar

lines overexpressing *UGT72B37* [49] were analyzed but no differences in the lignin content and the saccharification potential were observed compared to the WT (Figure S2).

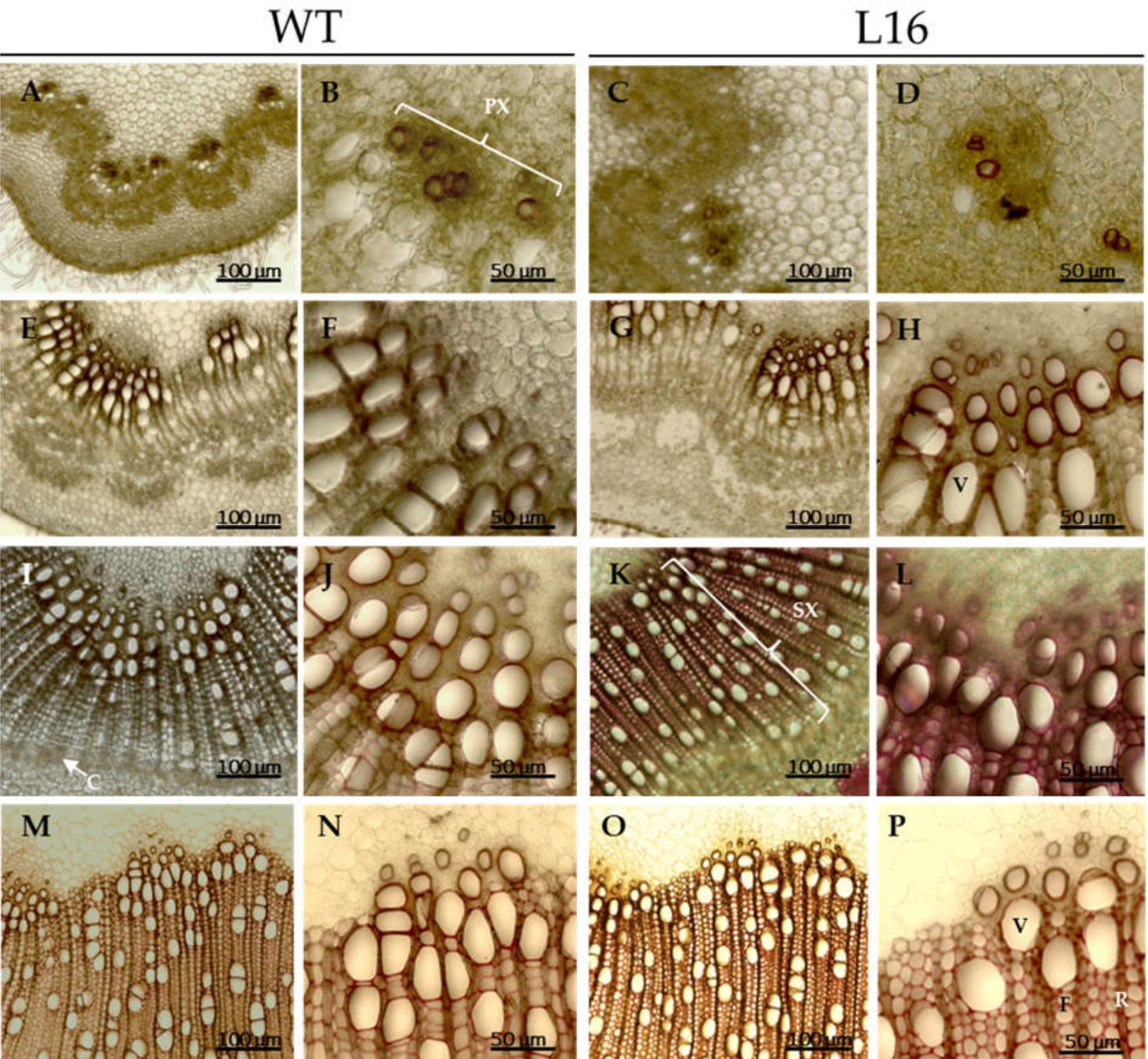

**Figure 4.** Wiesner staining of cross sections of stems of three-month-old WT and L16 *ugt72b37* mutants: (**A–D**) 1 cm below the apex; (**E–H**), 10 cm below the apex; (**I–L**), 20 cm below the apex; and (**M–P**), 30 cm below the apex. (**B,D,F,H,J,L,N,P**) were close-ups of (**A,C,E,G,I,K,M,O**) respectively. Lignin is evidenced by pink staining. PX, primary xylem; V, vessel; C, cambium; SX, secondary xylem; F, fiber; R, ray.

### 3.4. Expression Level of Genes Involved in Monolignol Biosynthesis and Polymerization

The expression level of several genes involved in lignin biosynthesis and polymerization was analyzed for *ugt72b37* mutant lines, in stem sections where secondary xylem formation was ongoing (between 20 and 25 cm below the apex). Because of the homogenous phenotype and the impact observed for the three lines on lignin content and composition, cellulose content, and potential of saccharification, this analysis was performed only on L6 and L16, which are the two lines with adequate biological replicates. Although the expression of several genes, such as *PHENYLALANINE AMMONIA-LYASE2 (PAL2)*, *CAFFEOYL-*

*COA O-METHYLTRANSFERASE1-2 (CCoAOMT1-2), FERULATE 5-HYDROXYLASE 1 (F5H1)* and *CAFFEIC ACID O-METHYLTRANSFERASE 2 (COMT2)*, was higher compared to the WT, this increase was moderate (maximum 1.5-fold) and not homogeneous for both mutants (Figure 5). There was no significant difference in the expression of *LAC4, LAC17*, and *PRX72* monolignol polymerization genes.

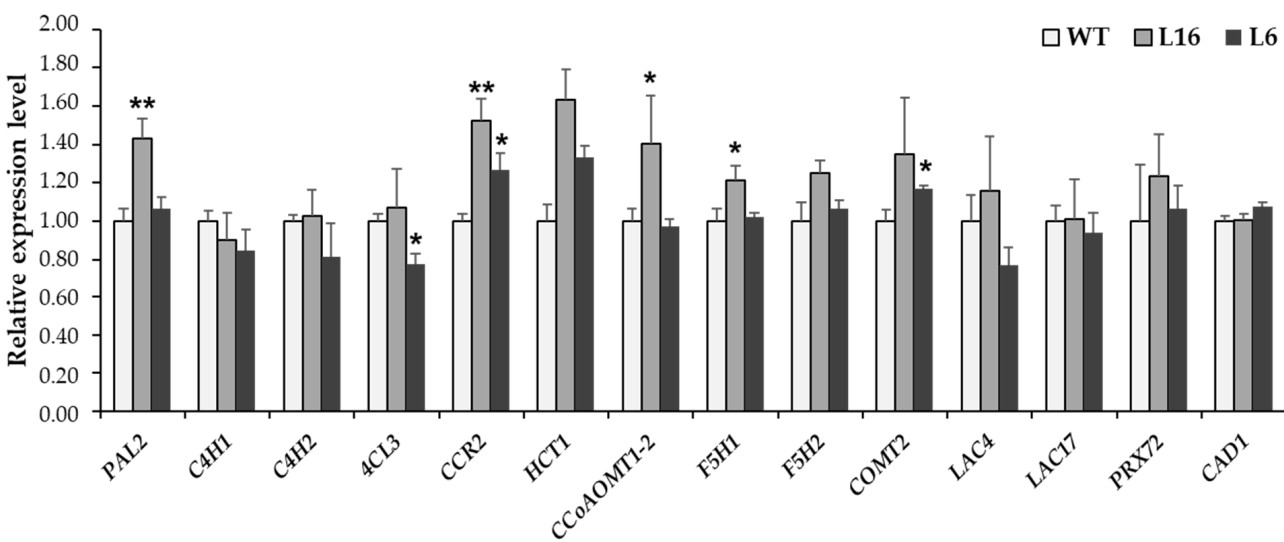

**Figure 5.** Relative expression level of poplar genes involved in lignin biosynthesis and polymerization in the xylem of stems of three-month-old *ugt72b37* mutant (L16 and L6) and WT lines cultivated in soil, as assessed by RT-qPCR. The expression level of each gene in each sample was normalized using two reference genes, *CDC2* and *CYC063*. The expression level of each gene in the WT was defined as the reference level, with a relative value = 1. Values are means of four biological replicates (±SE). *: significant difference ($p < 0.05$) compared with the WT, **: significant difference ($p < 0.01$) compared with the WT (*t*-test).

*CCR2* was the only tested gene with an expression significantly upregulated in both lines (by 1.2 and 1.5-fold in L16 and L6, respectively). This gene codes for CINNAMOYL-COA REDUCTASE 2, the enzyme catalyzing the first specific step in monolignol biosynthesis. The increase of its transcript level might be linked to a biological impact on the observed lignin accumulation (Figure 3).

### 3.5. Impact of the ugt72b37 Mutation on the Expression of the Other Poplar UGT72B Genes

The alteration of the *UGT72B37* expression may affect the expression level of other members of the poplar *UGT72B* family, which is made of four members, *UG72B36-B39* [49]. To evaluate this hypothesis, the expression of *UGT72B36, UGT72B38* and *UGT72B39* was measured in stems of the L16 and L6 *ugt72b37* mutants and compared to the WT (Figure 6). Whereas the expression of *UGT72B36* (downregulated in L6) and *UGT72B39* (overexpressed in L16) was slightly altered compared to the WT, the expression of *UGT72B38* was increased by 2.3- and 2.4-fold in L6 and L16, respectively, suggesting a functional compensation of the mutation impact by another member of the gene family, as it was suggested for the *ugt72b1 Arabidopsis* mutant [41].

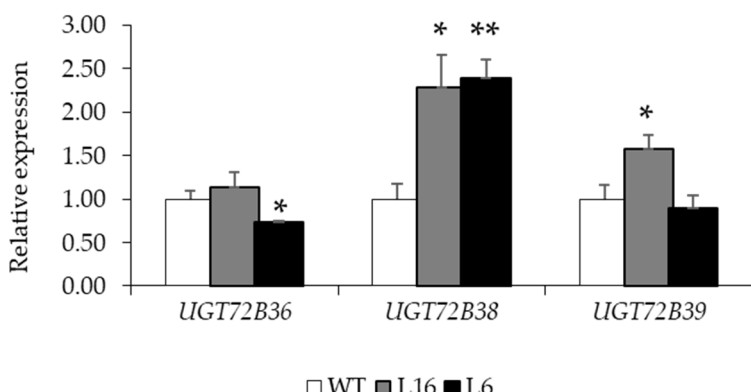

**Figure 6.** Expression level of the 3 other poplar *UGT72Bs* in the xylem of three-month-old *ugt72b37* mutant and WT lines cultivated in soil, as assessed by RT-qPCR. The expression level of each gene in WT was defined as reference level, with a relative value = 1. Values are means of four biological replicates (±SE). *: significative difference ($p < 0.05$) compared with the WT, **: significative difference ($p < 0.01$) compared with the WT (*t*-Test).

## 4. Discussion

### 4.1. UGT72B37 Has a Role in the Xylem Lignification Process

In the present study, the characterization of three-month-old *ugt72b37* poplar mutants revealed three main outcomes, when compared to the WT.

First, the mutation of the *UGT72B37* triggered a 10% increase in the lignin content of the secondary xylem (Figure 3) without ectopic lignification (Figure 4). Similarly to UGT72B37, the *P. bretschneideri* UGT72AJ2 glucosylates coniferyl alcohol and sinapyl alcohol [48]. Its coding gene is expressed during the early stages of fruit development, mainly in lignifying tissues such as stone cells. Transient silencing of its expression in the fruit through RNA interference resulted in a slightly higher Klason lignin content, as well as in an increase in the stone cell content, suggesting that PbUGT72AJ2 not only affects lignification, but also the developmental pattern of the fruits. Besides, several *LACs* and *PRXs* were more expressed in the silenced fruits than in the WT [48], pointing to a possible higher monolignol polymerization capacity. In addition, transient overexpression of *PbUGT72AJ2* did not modify the lignin content, the stone cell content, nor the expression of *LACs* and *PRXs* [48]. This result is consistent with the absence of a noticeable phenotype in the transgenic poplar lines overexpressing *UGT72B37*.

Previously, the characterization of *Arabidopsis* lines with altered expression (overexpression or downregulation) of *UGT72E1-3* did not reveal a change in lignin content or composition, except for the *ugt72e3* mutant [37,38,67]. When compared to the WT, no difference in lignin content was evidenced in the *ugt72e3* mutant in both young and old parts of inflorescence stems when measured by the acetyl bromide procedure. However, a 40% increase in lignin content in the cell walls of both xylem and interfascicular fibers was demonstrated by Raman micro-spectroscopy and safranin O ratiometric imaging technique in the young part (but not in the old part) from the inflorescence stem of the *ugt72e3* mutant [67]. This result was correlated with a higher incorporation capacity of fluorescently labeled monolignols into the lignin in interfascicular fibers (by 1.5-fold for H units, ~2-fold for G units, and ~2.5-fold for S units) and xylem (by ~2.5-fold for G units, and 2.2-fold for S units) of the young stem of the mutant compared to the WT, as analyzed on cross-sections. The higher incorporation of monolignols was associated with the increased expression level of the *PRX71* (by ~4-fold) and the *LAC17* (by ~2.2-fold) genes. As hypothesized by these authors, the increased monolignol flux, resulting from the lower glycosylation level, may induce the expression of polymerization genes. In contrast with the results described in [67], in the present study, no differences in the expression of genes involved in lignin polymerization were noticed in the *ugt72b37* poplar mutant (Figure 5). However, many other LACs and PRXs may partake in this process, both at transcriptional and post-

transcriptional levels (such as enzymatic activity, as reviewed in [68]). In this context, it would be relevant to evaluate the capacity of the *ugt72b37* mutant cell walls to incorporate H-, G-, and S-monolignol reporters. Besides, wood formation in perennial woody plants such as poplar is a dynamic process that is spatio-temporally regulated [69]. As the lignin content was measured in young poplar plants forming most probably earlywood, the increased lignin content may be the result of an earlier lignification of xylem cell walls in the mutant that might be overcome upon completion of the growth ring and latewood formation.

Second, albeit the monomer composition and yield, as analyzed by thioacidolysis, not being altered when considering the whole cell wall, a lower proportion of H, G and S units was measured in the lignin of the *ugt72b37* mutants compared to the WT. This outcome suggests that in the lignin of the *ugt72b37* mutants, more monolignols are linked by carbon–carbon (C-C) linkages that are not considered when using thioacidolysis (Figure S1). The impact of the mutation on the proportion of condensed/not condensed lignin may be due also to differences in the developmental stage of the cell walls that are possibly overwhelmed during successive growth stages. Alternatively, the monolignol transport rate is possibly modified in the *ugt72b37* mutant due to the increased availability of aglycone monolignols. As demonstrated in [70] by in silico lignin biosynthesis simulations, once the monolignol transport rate increases, more β-β (C-C) and fewer β-O-4 (C-O) linkages are formed resulting in a reduction of the monomers units involved in non-condensed lignin.

Third, in contrast with the mutation of *UGT72B37*, the overexpression of this gene in transgenic poplar did not lead to a modification in lignin content nor in saccharification efficiency (Figure S2) [49]. As *UGT72B37* is already highly expressed in the stem, there is possibly little impact of its overexpression on the lignification process [49]. UGT72B37 may be already present in excess or the increase in monolignol glucosides is compensated by other processes such as glucoside hydrolysis catalyzed by *β*-glucosidases or incorporation of the glucosides in the growing lignin polymer.

### 4.2. The UGT72B37 Mutation in Poplar Has a Different Impact Than the UGT72B1 Mutation in Arabidopsis and the LuUGT175 Mutation in Flax

No specific phenotype was observed and the overall plant growth of the *ugt72b37* poplar mutants was not altered in this perennial species. In *Arabidopsis*, the mutation of *UGT72B1* [41], belonging to the same sub-family as *UGT72B37*, triggers contrasting consequences compared with our results such as (i) an ectopic lignification, as shown by Wiesner and Mäule staining of cross sections of inflorescence stems, (ii) an alteration of the non-condensed lignin composition characterized by a ~four-fold increase in the S/G ratio and the release of 60% more monomers, mostly S units, as analyzed also by thioacidolysis, (iii) a ~three-fold increase in the amount of coniferin in stems and (iv) an arrest of growth, reduction in fertility and pigment deposition as well as three times thicker pith cell walls. Although the lignin content was not quantified, the growth arrest and the fertility problems may be caused by the enhanced and ectopic lignification in the floral stem of the *ugt72b1* mutants, in the organs where the gene was normally expressed, as suggested in [41].

Like the *ugt72b1 Arabidopsis* mutant, stem section colorations (Wiesner and Mäule stainings) of the *luugt175* flax mutant evidenced an ectopic lignification which was observed in phloem fibers. In addition, the xylem portion was much higher in the mutant than in the wild type and the stem was characterized by a higher amount of H, G, and S monomers content, as measured by thioacidolysis [51]. Furthermore, a blockage of growth and development leading to 11%–25% shorter plants was observed in the *luugt715* mutant compared to the WT.

The discrepancies between our results and the results obtained for the *ugt72b1* [41] and the *luugt715* [51] mutants suggest either that poplar *UGT72B37* is not orthologous to *UGT72B1* and *LuUGT715* or that the mutation of these genes triggers a more severe pheno­type than that of *UGT72B37* because of redundancy of gene function, such as *UGT72B38* which expression was found to be induced in the *ugt72b37* mutants (Figure 6). A comple-

mentation study of the *ugt72b1* and/or the *luugt715* mutants with *UGT72B37*, *UGT72B38*, or both may help to verify this hypothesis. Another hypothesis is that both *Arabidopsis* and flax are annual species, and the observed differences may be explained by the very different developmental pattern of poplar as perennial woody species. Alternatively, the differences in the observed phenotypes, such as the ectopic lignification and the resulting impact on growth in the *ugt72b1* and the *luugt715* mutants but not in the *ugt72b37* mutants, arise from the different patterns of expression of *UGT72B37*, *UGT72B1*, and *LuUGT715*. Previously, we showed that the poplar *UGT72B37* was mainly expressed within the secondary xylem compared to other plant organs and tissues [49]. In *Arabidopsis*, *UGT72B1* expression was localized mainly within the xylem, the cortex, and the pith of the younger part of the floral stem, as analyzed by promoter-*GUS* fusion. In the older part of the stem, *GUS* expression was found only in the primary xylem whereas the expression in the secondary xylem was not investigated [41]. In flax, *LuUGT715* is expressed in stem and fibers, based on public RNA-seq data [51].

Ectopic lignification or increased lignin content by overexpressing or ectopically expressing genes involved in lignin biosynthesis or its regulation have been reported in many studies (reviewed in [71]) and may cause morphological defects. As an increased lignin content may be associated with other variations in the cell wall, such as in the cellulose content or the lignin composition [72,73], it is difficult to evaluate the impact of lignin content alteration alone. However, in other cases, the increase in lignin content does not affect the overall plant growth, such as the overexpression of the *Arabidopsis AtMYB55*, *AtMYB61*, or *AtMYB63* in rice that led to up to 53% increase in lignin content [74].

### 4.3. The Increase in Lignin Content Measured in the ugt72b37 Mutants May Be Linked to an Upregulation of CCR2

In the *ugt72b37* mutants, the analysis of the expression level of genes involved in monolignol biosynthesis or polymerization did not evidence consistent changes compared to the WT, indicating that the higher lignin content measured in these mutants is not strictly regulated at a transcriptional level (Figure 5). However, a low but significant increase in the expression of *CCR2*, which is considered a key gene in lignin biosynthesis [75], may be responsible for the production of a higher amount of monolignols. In accordance, the overexpression of *Brassica napus Bnac.CCR2.b* gene was found to induce an 11%–13% increased lignin content in the stems of *B. napus,* without affecting agronomic traits [76].

Opposite to our results, and as analyzed by RNA-seq and RT-qPCR, enhanced transcription of lignification-related genes including monolignol biosynthesis, transport, and polymerization, as well as genes involved in the regulation of secondary cell wall formation was reported in the *Arabidopsis ugt72b1* mutants [41]. Other modifications were noticed, such as for genes involved in carbohydrate metabolism, stress adaptation, and development. It was hypothesized that the depletion of monolignol glucosides observed in the *ugt72b1* mutants triggers a signal upregulating genes involved in the lignin biosynthesis pathway as well as in lignin polymerization, inducing an increase in the production of monolignols and a subsequent ectopic lignification [41].

The analysis of the expression of the other members of the *UGT72B* subfamily in the *ugt72b37* poplar mutants evidenced a 2.3–2.4-fold increase in the expression of *UGT72B38* (Figure 6). As this gene is the closest homolog to *UGT72B37* [49], there is possibly a compensation by *UGT72B38*. Similarly, in *Arabidopsis*, the *ugt72b1* mutation was compensated by an upregulation of both *UGT72B3* and *UGT72E2* in the mutant lines [41]. There are no data on the substrate specificity of the enzyme coded by *UGT72B38*, but it would be relevant to generate double *ugt72b37 ugt72b38* poplar mutants to override this putative compensation and evaluate the impact of this mutation on lignin content.

Overall, the increase in monolignol flux to the cell wall, due to the reduction in monolignol glycosylation, might be considered as the main cause of the increased lignin content. Upon synthesis, monolignols are immediately transported to the cell wall where they are polymerized.

## 5. Conclusions

Here, UGT72B37 was found to play a role in the lignification of poplar xylem, possibly by regulating monolignols flux towards the cell wall where they are incorporated into the lignin polymer. However, as the observations were made in three-month-old trees and the anatomical traits, as well as lignin content and composition, may evolve during xylem development, it would be very informative to follow these features over successive years in natural environmental conditions.

In addition, lignin is a major player in the response of plants to various stresses [77]. Previous studies have reported that increasing lignin content could protect plants from the invasion and spread of pathogens as demonstrated for *B. napus* overexpressing *BnaC.CCR2.b* [76]. Therefore, the impact of the increased lignin content on the response of *ugt72b37* poplar mutants to biotic as well as abiotic stresses is also worth investigating.

**Supplementary Materials:** The following supporting information can be downloaded at: https://www.mdpi.com/article/10.3390/f13122167/s1, Figure S1. Cell wall components and saccharification efficiency in stems of three-month-old *ugt72b37* poplar mutants and WT. Figure S2. Characterization of 4-month-old transgenic poplar lines overexpressing *UGT72B37* and WT. Table S1. List of primers (5′-3′) used in this study. Table S2. Phenotypic parameters and associated standard error (SE) of 14-week-old mutant and WT lines.

**Author Contributions:** Conceptualization, H.A.H., M.B. (Marc Behr), M.B. (Marie Baucher) and M.E.J.; software, H.A.H., M.B. (Marc Behr) and C.G.; formal analysis, H.A.H., R.S., M.B. (Marc Behr), M.B. (Marie Baucher) and C.G.; investigation, H.A.H., M.B. (Marc Behr), C.G., R.S. and A.M.; writing—original draft preparation, M.B. (Marie Baucher), H.A.H., C.G. and M.E.J.; writing—review and editing, M.B. (Marc Behr), M.B. (Marie Baucher), M.B. (Moussa Baragé), M.E.J., R.S. and C.G.; visualization, H.A.H., M.B. (Marc Behr), C.G. and R.S.; supervision, M.B. (Marc Behr), C.G., M.E.J. and M.B. (Marie Baucher); funding acquisition, M.B. (Marie Baucher). All authors have read and agreed to the published version of the manuscript.

**Funding:** M. Behr and C.G. are supported by Belgian Fonds de la Recherche Scientifique (FRS-FNRS) research projects T.0068.18 and T.0010.22, respectively. M. Baucher is a Senior Research Associate of the FRS-FNRS. H.A.H. was the recipient of pre-doctoral grants from the Académie de Recherche et d'Enseignement Supérieur, Belgium (ARES).

**Data Availability Statement:** Not applicable.

**Acknowledgments:** The authors acknowledge Eric Givron for his valuable assistance in lignin analysis and Véronique Megalizzi for her help in protein modeling.

**Conflicts of Interest:** The authors declare no conflict of interest.

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
