# Peer review of "A Higher Lignin Content in ugt72b37 Poplar Mutants Indicates a Role of Monolignol Glycosylation in Xylem Lignification"

_forests, doi:10.3390/f13122167_

Round 1

Reviewer 1 Report

I have a few major remarks.

Line 222: First, the mutation of the UGT72B37 was found to trigger a 10% increase of lignin content in the secondary xylem (Figure 3) without ectopic lignification (Figure 4).

This study correctly demonstrates this for 3-month plants. However, the growth rings were still not completed at 3 months. The microscopy shows developing xylem. Apparently, the xylem fibers appear stained in the genetically modified plants, suggesting that the xylem cell development is advanced compared to the WT. But it doesn’t mean that the WT will not catch up later in development – the xylem fibers will normally become lignified. So, complete growth rings must be compared to claim that the mutation causes increased lignification.

Therefore, the discussion and conclusion of this manuscript need to be amended.

Line 224: Albeit the monomer composition and yield, as analyzed by thioacidolysis, were not altered when considering the whole cell wall, a lower proportion of H, G and S units was measured in the lignin of the ugt72b37 mutants compared to 226 the WT.

Again, you must be careful regarding the proportions of H, G, and S units when comparing vessels and fibers versus only vessels in developing xylem.

When comparing poplar with Arabidopsis, it should be considered that poplar is a tree with continuous secondary growth.

Line: 162: In parallel, 4-month-old transgenic poplar lines overexpressing UGT72B37 [45] were analyzed but no differences in the lignin content and the saccharification potential were observed compared to the WT (Figure S3).

I was confused because this is not in the Methods nor Discussed later in the paper.

Reviewer 2 Report

It is apparent that this was a multi-disciplinary effort involving a lot of work. I think it could potentially be of great interest, so I hope this manuscript will achieve publication status. However, some problems exist in the present version.  

In general, the citations in the Introduction are somewhat misleading, ignoring highly relevant research published before the year 2000 (excepting refs. 24, 27) and taking liberties with what was actually described in publications that are cited.  For example, lines 46,47 state "coniferin, have been found to accumulate in cambial tissue and differentiating xylem, mostly in conifers [21–24]" but the work that actually showed accumulation was earlier and is uncited.  By overlooking the foundational contributions, the manuscript conveys weak scholarship, objectivity and respect for those upon whose shoulders present investigations stand. 

Lines 55-81 of the Introduction would be better positioned as Discussion.

Much of lines 90-134 appear to be Methods rather than Results.

In order for readers to appreciate if real differences existed between WT and mutants in lignin content, monolignols, etc., it is important to provide a table of morphological data and photographs of the actual trees that were investigated.  The height, 30-cm stem diameter, leaf number and size, and stem orientation all need to be revealed, as it is well established that these are variables affecting lignification biochemistry.  For example, the finding reported on lines 143, 144 of "a significant increase of ~10 % in lignin content was observed in the three mutant lines comparatively with the WT" could be explainable in terms of difference in phenotype.  There is a need to convince the reader otherwise by providing data and photographs.

Concerning the description of the analysis of monolignol glucosides (lines 370 – 376), methods of Speeckaert et al. (2020) were used, several of the authors of this manuscript also authored that paper.  Inexplicably, Figure S1 of this manuscript showing “HPLC-UV profiles of phenolic compounds extracted from wood samples of debarked stems of WT and ugt72b37 mutant lines” does not agree with Speeckaert et al. (2020).   Speeckaert et al. (2020) described coniferin and other glucosides eluting around 9 minutes and the aglycones about a minute later, whereas Figure S1 states “Retention time of coniferin, 2.75 min; syringin, 3.39 min; coniferyl alcohol, 7.43 min; sinapyl alcohol, 10.94 min; coniferaldehyde, 11.3 min; sinapaldehyde, 11.54 min.”  On the other hand, lines 138, 139 state "coniferin and syringin were not detected in any of the samples (Figure S1)."  So, this is very confusing!  Perhaps citing Speeckaert et al. (2020) does not provide an adequate description of what was actually done, or perhaps this manuscript contains a fundamental analytical error (in which case it should be withdrawn and the work redone), or perhaps it is simply a problem in communications.  Major revision may be needed because of the attending uncertainty.

Related to the foregoing, the methods as stated in the manuscript and in Figure S1 do not make clear what tissue(s) was actually investigated.  To my reading, only mature poplar xylem - not cambium or developing xylem - was investigated (see lines 138, 139).  But if so, why was the region of wood formation not investigated, when ample past work has reported the monolignol glucosides to exist in the region of xylem development?

    ref 4: no date

I wonder if the authors are aware that there is published evidence in poplar wood for the existence of monolignols in addition to those implicated in H, G and S lignins? Thorough scholarship is needed.

Reviewer 3 Report

Nowadays there are studies that suggest the existence of xylogenesis markers, various cytological, biochemical, and molecular genetic indicators that could identify wood quality. Scientists need to look for the ways that will help to control the processes underlying xylogenesis to increase plant productivity. Without a doubt, the present study is a very high-level study, and the results presented by the authors are relevant. Of course in terms of its content the paper fits into the scope of Plants journal. The results have both fundamental and practical significance. Materials and Methods are described in detail and match the aims completely.

Minor points

1.      English language editing

Line 141 – is should be replaced to was

Line 142 – is should be replaced to was

Line 161 – are should be replaced to were

Line 178 – are should be replaced to were

Line 182 – are should be replaced to were

Line 243 – is should be replaced to was

Line 246 – is should be replaced to was

Line 254 – is should be replaced to was

Line 270 – is should be replaced to was

2.      Arabidopsis should be written in Italics

3.      Figures

Line 166 – Figure 3 – the letter under L 11 is absent because of a low number of biological replicates. Are technical replicates included in the total SE for all the variants?

Figure S2. You don’t need to mention p> 0.05.

Figure S2 А CASA lignin content – Please, replace commas with dots in values.

Figure S2 В – Could you please start the y-axis with, for example, 20 or 25 percent, otherwise the bars are not visible.

Figure S2 С – Could you please start the y-axis with, for example, 400 µg/mg, otherwise the bars are not visible.

Figure S2 В and С – Can the T-test be used for such a small number of biological replicates? Was the raw data tested for normality?

Round 2

Reviewer 1 Report

After the revision, Figure 4 is considerably improved, showing that the development of the two lines is similar. 
I would point out that the growth rings are not completely developed.

Reviewer 2 Report

At L 404 it reads: Previously, we showed that the poplar UGT72B37 was mainly expressed within the stem secondary xylem [49]. If that is correct, if greater expression occurs within the secondary xylem than within the cambium, it would seem to me that you have there a highly important observation that should be emphasized and discussed.

The phloroglucinol HCl lignin response is usually red, unlike what your Figure 4 shows.  I suggest you address this difference in the results or discussion, or provide a revised figure.
